# Twins: Revisiting the Design of Spatial Attention in Vision Transformers

**Xiangxiang Chu**[1], **Zhi Tian**[1,2], **Yuqing Wang**[1], **Bo Zhang**[1]
**Haibing Ren**[1], **Xiaolin Wei**[1], **Huaxia Xia**[1], **Chunhua Shen**[2*]

[1] Meituan Inc.      [2] The University of Adelaide, Australia

[1] {chuxiangxiang,wangyuqing06,zhangbo97,renhaibing,weixiaolin02,xiahuaxia}@meituan.com
[2] zhi.tian@outlook.com, chunhua@me.com

## Abstract

Very recently, a variety of vision transformer architectures for dense prediction tasks have been proposed and they show that the design of spatial attention is critical to their success in these tasks. In this work, we revisit the design of the spatial attention and demonstrate that a carefully devised yet simple spatial attention mechanism performs favorably against the state-of-the-art schemes. As a result, we propose two vision transformer architectures, namely, Twins-PCPVT and Twins-SVT. Our proposed architectures are highly efficient and easy to implement, only involving matrix multiplications that are highly optimized in modern deep learning frameworks. More importantly, the proposed architectures achieve excellent performance on a wide range of visual tasks including image-level classification as well as dense detection and segmentation. The simplicity and strong performance suggest that our proposed architectures may serve as stronger backbones for many vision tasks. Our code is available at: https://git.io/Twins.

## 1 Introduction

Recently, Vision Transformers [1–3] have received increasing research interest. Compared to the widely-used convolutional neural networks (CNNs) in visual perception, Vision Transformers enjoy great flexibility in modeling long-range dependencies in vision tasks, introduce less inductive bias, and can naturally process multi-modality input data including images, videos, texts, speech signals, and point clouds. Thus, they have been considered to be a strong alternative to CNNs. It is expected that vision transformers are likely to replace CNNs and serve as the most basic component in the next-generation visual perception systems.

One of the prominent problems when applying transformers to vision tasks is the heavy computational complexity incurred by the spatial self-attention operation in transformers, which grows quadratically in the number of pixels of the input image. A workaround is the *locally-grouped self-attention* (or self-attention in non-overlapped windows as in the recent Swin Transformer [4]), where the input is spatially grouped into non-overlapped windows and the standard self-attention is computed only within each sub-window. Although it can significantly reduce the complexity, it lacks the connections between different windows and thus results in a limited receptive field. As pointed out by many previous works [5–7], a sufficiently large receptive field is crucial to the performance, particularly for dense prediction tasks such as image segmentation and object detection. Swin [4] proposes a shifted window operation to tackle the issue, where the boundaries of these local windows are gradually moved as the network proceeds. Despite being effective, the shifted windows may have uneven sizes. The uneven windows result in difficulties when the models are deployed with ONNX or TensorRT,

---

*Corresponding author.

35th Conference on Neural Information Processing Systems (NeurIPS 2021).

which prefers the windows of equal sizes. Another solution is proposed in PVT [8]. Unlike the standard self-attention operation, where each query computes the attention weights with all the input tokens, in PVT, each query only computes the attention with a sub-sampled version of the input tokens. Although its computational complexity in theory is still quadratic, it is already manageable in practice.

From a unified perspective, the core in the aforementioned vision transformers is how the spatial attention is designed. Thus, in this work, we revisit the design of the spatial attention in vision transformers. Our first finding is that the global sub-sampled attention in PVT is highly effective, and with the applicable positional encodings [9], its performance can be on par or even better than state-of-the-art vision transformers (*e.g.*, Swin). This results in our first proposed architecture, termed *Twins-PCPVT*. On top of that, we further propose a *carefully-designed yet simple spatial attention* mechanism, making our architectures more efficient than PVT. Our attention mechanism is inspired by the widely-used separable depthwise convolutions and thus we name it *spatially separable self-attention* (SSSA). Our proposed SSSA is composed of two types of attention operations—(i) *locally-grouped self-attention* (LSA), and (ii) *global sub-sampled attention* (GSA), where LSA captures the fine-grained and short-distance information and GSA deals with the long-distance and global information. This leads to the second proposed vision transformer architecture, termed *Twins-SVT*. It is worth noting that both attention operations in the architecture are *efficient and easy-to-implement* with matrix multiplications in a few lines of code. Thus, all of our architectures here have great applicability and can be easily deployed.

We benchmark our proposed architectures on a number of visual tasks, ranging from image-level classification to pixel-level semantic/instance segmentation and object detection. Extensive experiments show that both of our proposed architectures perform favorably against other state-of-the-art vision transformers with similar or even reduced computational complexity.

## 2   Related Work

**Convolutional neural networks.**   Characterized by local connectivity, weight sharing, shift-invariance and pooling, CNNs have been the *de facto* standard model for computer vision tasks. The top-performing models [10–13] in image classification also serve as the strong backbones for downstream detection and segmentation tasks.

**Vision Transformers.** Transformer was firstly proposed by [14] for machine translation tasks, and since then they have become the state-of-the-art models for NLP tasks, overtaking the sequence-to-sequence approach built on LSTM. Its core component is multi-head self-attention which models the relationship between input tokens and shows great flexibility.

In 2020, Transformer was introduced to computer vision for image and video processing [1–3, 9, 15–17, 17–32]. In the image classification task, ViT [1] and DeiT [2] divide the images into patch embedding sequences and feed them into the standard transformers. Although vision transformers have been proved compelling in image classification compared with CNNs, a challenge remains when it is applied to dense prediction tasks such as object detection and segmentation. These tasks often require feature pyramids for better processing objects of different scales, and take as inputs the high-resolution images, which significantly increase the computational complexity of the self-attention operations.

Recently, Pyramid Vision Transformer (PVT) [8] is proposed and can output the feature pyramid [33] as in CNNs. PVT has demonstrated good performance in a number of dense prediction tasks. The recent Swin Transformer [4] introduces non-overlapping window partitions and restricts self-attention within each local window, resulting in linear computational complexity in the number of input tokens. To interchange information among different local areas, its window partitions are particularly designed to shift between two adjacent self-attention layers. The semantic segmentation framework OCNet [34] shares some similarities with us and they also interleave the local and global attention. Here, we demonstrate this is a general design paradigm in vision transformer backbones rather than merely an incremental module in semantic segmentation.

**Grouped and Separable Convolutions.** Grouped convolutions are originally proposed in AlexNet [35] for distributed computing. They were proved both efficient and effective in speeding up the networks. As an extreme case, depthwise convolutions [12, 36] use the number of groups that is

equal to the input or output channels, which is followed by point-wise convolutions to aggregate the information across different channels. Here, the proposed spatially separable self-attention shares some similarities with them.

**Positional Encodings.** Most vision transformers use absolute/relative positional encodings, depending on downstream tasks, which are based on sinusoidal functions [14] or learnable [1, 2]. In CPVT [9], the authors propose the conditional positional encodings, which are dynamically conditioned on the inputs and show better performance than the absolute and relative ones.

## 3  Our Method: Twins

We present two simple yet powerful spatial designs for vision transformers. The first method is built upon PVT [8] and CPVT [9], which only uses the global attention. The architecture is thus termed Twins-PCPVT. The second one, termed Twins-SVT, is based on the proposed SSSA which interleaves local and global attention.

### 3.1  Twins-PCPVT

PVT [8] introduces the pyramid multi-stage design to better tackle dense prediction tasks such as object detection and semantic segmentation. It inherits the absolute positional encoding designed in ViT [1] and DeiT [2]. All layers utilize the global attention mechanism and rely on spatial reduction to cut down the computation cost of processing the whole sequence. It is surprising to see that the recently-proposed Swin transformer [4], which is based on shifted local windows, can perform considerably better than PVT, even on dense prediction tasks where a sufficiently large receptive field is even more crucial to good performance.

In this work, we surprisingly found that the less favored performance of PVT is mainly due to the *absolute positional encodings* employed in PVT [8]. As shown in CPVT [9], the absolute positional encoding encounter difficulties in processing the inputs with varying sizes (which are common in dense prediction tasks). Moreover, this positional encoding also breaks the *translation invariance*. On the contrary, Swin transformer makes use of the relative positional encodings, which bypasses the above issues. Here, we demonstrate that this is the main cause why Swin outperforms PVT, and we show that if the appropriate positional encodings are used, PVT can actually achieve on par or even better performance than the Swin transformer.

Here, we use the conditional position encoding (CPE) proposed in CPVT [9] to replace the absolute PE in PVT. CPE is conditioned on the inputs and can naturally avoid the above issues of the absolute encodings. The position encoding generator (PEG) [9], which generates the CPE, is placed after the first encoder block of each stage. We use the simplest form of PEG, *i.e.*, a 2D depth-wise convolution without batch normalization. For image-level classification, following CPVT, we remove the class token and use global average pooling (GAP) at the end of the stage [9]. For other vision tasks, we follow the design of PVT. Twins-PCPVT inherits the advantages of both PVT and CPVT, which makes it easy to be implemented efficiently. Our extensive experimental results show that this simple design can match the performance of the recent state-of-the-art Swin transformer. We have also attempted to replace the relative PE with CPE in Swin, which however does not result in noticeable performance gains, as shown in our experiments. We conjecture that this maybe due to the use of shifted windows in Swin, which might not work well with CPE.

**Architecture settings**  We report the detailed settings of Twins-PCPVT in Table 2 (in supplementary), which are similar to PVT [8]. Therefore, Twins-PCPVT has similar FLOPs and number of parameters to [8].

### 3.2  Twins-SVT

Vision transformers suffer severely from the heavy computational complexity in dense prediction tasks due to high-resolution inputs. Given an input of $H \times W$ resolution, the complexity of self-attention with dimension $d$ is $\mathcal{O}(H^2W^2d)$. Here, we propose the spatially separable self-attention (SSSA) to alleviate this challenge. SSSA is composed of locally-grouped self-attention (LSA) and global sub-sampled attention (GSA).

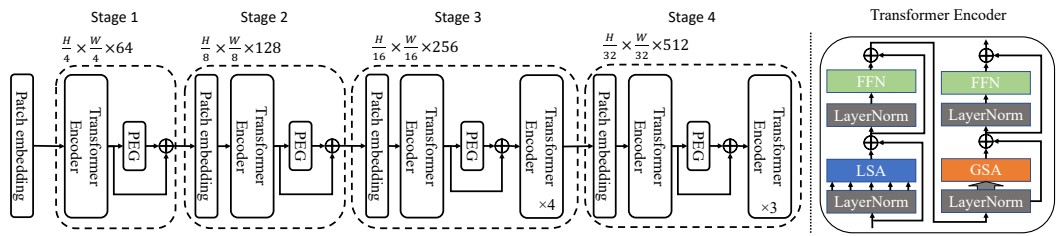

**Figure 1** – Architecture of Twins-SVT-S. "PEG" is the positional encoding generator from CPVT [9].

**Locally-grouped self-attention (LSA).** Motivated by the group design in depthwise convolutions for efficient inference, we first equally divide the 2D feature maps into sub-windows, making self-attention communications only happen within each sub-window. This design also resonates with the multi-head design in self-attention, where the communications only occur within the channels of the same head. To be specific, the feature maps are divided into $m \times n$ sub-windows. Without loss of generality, we assume $H\%m = 0$ and $W\%n = 0$. Each group contains $\frac{HW}{mn}$ elements, and thus the computation cost of the self-attention in this window is $\mathcal{O}(\frac{H^2W^2}{m^2n^2}d)$, and the total cost is $\mathcal{O}(\frac{H^2W^2}{mn}d)$. If we let $k_1 = \frac{H}{m}$ and $k_2 = \frac{W}{n}$, the cost can be computed as $\mathcal{O}(k_1 k_2 HWd)$, which is significantly more efficient when $k_1 \ll H$ and $k_2 \ll W$ and grows linearly with $HW$ if $k_1$ and $k_2$ are fixed.

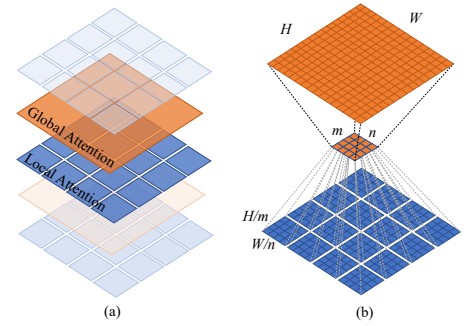

(a)  (b)

**Figure 2** – **(a)** Twins-SVT interleaves locally-grouped attention (LSA) and global sub-sampled attention (GSA). **(b)** Schematic view of the locally-grouped attention (LSA) and global sub-sampled attention (GSA).

Although the locally-grouped self-attention mechanism is computation friendly, the image is divided into non-overlapping sub-windows. Thus, we need a mechanism to communicate between different sub-windows, as in Swin. Otherwise, the information would be limited to be processed locally, which makes the receptive field small and significantly degrades the performance as shown in our experiments. This resembles the fact that we cannot replace all standard convolutions by depth-wise convolutions in CNNs.

**Global sub-sampled attention (GSA).** A simple solution is to add extra standard global self-attention layers after each local attention block, which can enable cross-group information exchange. However, this approach would come with the computation complexity of $\mathcal{O}(H^2W^2d)$.

Here, we use a single representative to summarize the important information for each of $m \times n$ sub-windows and the representative is used to communicate with other sub-windows (serving as the key in self-attention), which can dramatically reduce the cost to $\mathcal{O}(mnHWd) = \mathcal{O}(\frac{H^2W^2d}{k_1k_2})$. This is essentially equivalent to using the sub-sampled feature maps as the key in attention operations, and thus we term it global sub-sampled attention (GSA). If we alternatively use the aforementioned LSA and GSA like separable convolutions (depth-wise + point-wise). The total computation cost is $\mathcal{O}(\frac{H^2W^2d}{k_1k_2} + k_1k_2HWd)$. We have $\frac{H^2W^2d}{k_1k_2} + k_1k_2HWd \geq 2HWd\sqrt{HW}$. The minimum is obtained when $k_1 \cdot k_2 = \sqrt{HW}$. We note that $H = W = 224$ is popular in classification. Without loss of generality, we use square sub-windows, $i.e.$, $k_1 = k_2$. Therefore, $k_1 = k_2 = 15$ is close to the global minimum for $H = W = 224$. However, our network is designed to include several stages with variable resolutions. Stage 1 has feature maps of $56 \times 56$, the minimum is obtained when $k_1 = k_2 = \sqrt{56} \approx 7$. Theoretically, we can calibrate optimal $k_1$ and $k_2$ for each of the stages. For simplicity, we use $k_1 = k_2 = 7$ everywhere. As for stages with lower resolutions, we control the summarizing window-size of GSA to avoid too small amount of generated keys. Specifically, we use the size of 4, 2 and 1 for the last three stages respectively.

As for the sub-sampling function, we investigate several options including average pooling, depth-wise strided convolutions, and regular strided convolutions. Empirical results show that regular strided convolutions perform best here. Formally, our spatially separable self-attention (SSSA) can be written as

$$
\begin{aligned}
\hat{\mathbf{z}}_{ij}^l &= \text{LSA}\left(\text{LayerNorm}\left(\mathbf{z}_{ij}^{l-1}\right)\right) + \mathbf{z}_{ij}^{l-1}, \\
\mathbf{z}_{ij}^l &= \text{FFN}\left(\text{LayerNorm}\left(\hat{\mathbf{z}}_{ij}^l\right)\right) + \hat{\mathbf{z}}_{ij}^l, \\
\hat{\mathbf{z}}^{l+1} &= \text{GSA}\left(\text{LayerNorm}\left(\mathbf{z}^l\right)\right) + \mathbf{z}^l, \\
\mathbf{z}^{l+1} &= \text{FFN}\left(\text{LayerNorm}\left(\hat{\mathbf{z}}^{l+1}\right)\right) + \hat{\mathbf{z}}^{l+1}, \\
i &\in \{1, 2, ...., m\}, j \in \{1, 2, ...., n\}
\end{aligned}
\tag{1}
$$

where LSA means locally-grouped self-attention within a sub-window; GSA is the global sub-sampled attention by interacting with the representative keys (generated by the sub-sampling functions) from each sub-window $\hat{\mathbf{z}}_{ij} \in \mathcal{R}^{k_1 \times k_2 \times C}$. Both LSA and GSA have multiple heads as in the standard self-attention. The PyTorch code of LSA is given in Algorithm 1 (in supplementary).

Again, we use the PEG of CPVT [9] to encode position information and process variable-length inputs on the fly. It is inserted after the first block in each stage.

**Model variants.** The detailed configure of Twins-SVT is shown in Table 3 (in supplementary). We try our best to use the similar settings as in Swin [4] to make sure that the good performance is due to the new design paradigm.

**Comparison with PVT.** PVT entirely utilizes global attentions as DeiT does while our method makes use of spatial separable-like design with LSA and GSA, which is more efficient.

**Comparison with Swin.** Swin utilizes the alternation of local window based attention where the window partitions in successive layers are shifted. This is used to introduce communication among different patches and to increase the receptive field. However, this procedure is relatively complicated and may not be optimized for speed on devices such as mobile devices. Swin Transformer depends on torch.roll() to perform cyclic shift and its reverse on features. This operation is memory unfriendly and rarely supported by popular inference frameworks such as NVIDIA TensorRT, Google Tensorflow-Lite, and Snapdragon Neural Processing Engine SDK (SNPE), etc. This hinders the deployment of Swin either on the server-side or on end devices in a production environment. In contrast, Twins models don't require such an operation and only involve matrix multiplications that are already optimized well in modern deep learning frameworks. Therefore, it can further benefit from the optimization in a production environment. For example, we converted Twins-SVT-S from PyTorch to TensorRT , and its throughput is boosted by $1.7\times$. Moreover, our local-global design can better exploit the global context, which is known to play an important role in many vision tasks.

Finally, one may note that the network configures (*e.g.*, such as depths, hidden dimensions, number of heads, and the expansion ratio of MLP) of our two variants are sightly different. This is intended because we want to make fair comparisons to the two recent well-known transformers PVT and Swin. PVT prefers a slimmer and deeper design while Swin is wider and shallower. This difference makes PVT have slower training than Swin. Twins-PCPVT is designed to compare with PVT and shows that a proper positional encoding design can greatly boost the performance and make it on par with recent state-of-the-art models like Swin. On the other hand, Twins-SVT demonstrates the potential of a new paradigm as to spatially separable self-attention is highly competitive to recent transformers.

## 4 Experiments

### 4.1 Classification on ImageNet-1K

We first present the ImageNet classification results with our proposed models. We carefully control the experiment settings to make fair comparisons against recent works [2, 8, 9]. All our models are trained for 300 epochs with a batch size of 1024 using the AdamW optimizer [37]. The learning rate is initialized to be 0.001 and decayed to zero within 300 epochs following the cosine strategy. We use a linear warm-up in the first five epochs and the same regularization setting as in [2]. Note that we do not utilize extra tricks in [26, 28] to make fair comparisons although it may further improve the

performance of our method. We use increasing stochastic depth [38] augmentation of 0.2, 0.3, 0.5 for small, base and large model respectively. Following Swin [4], we use gradient clipping with a max norm of 5.0 to stabilize the training process, which is especially important for the training of large models.

We report the classification results on ImageNet-1K [39] in Table 1. Twins-PCPVT-S outperforms PVT-small by $1.4\%$ and obtains similar result as Swin-T with 18% fewer FLOPs. Twins-SVT-S is better than Swin-T with about $35\%$ fewer FLOPs. Other models demonstrate similar advantages.

It is interesting to see that, without bells and whistles, Twins-PCPVT performs *on par* with the recent state-of-the-art Swin, which is based on much more sophisticated designs as mentioned above. Moreover, Twins-SVT also achieves similar or better results, compared to Swin, indicating that the spatial separable-like design is an effective and promising paradigm.

One may challenge our improvements are due to the use of the better positional encoding PEG. Thus, we also replace the relative PE in Swin-T with PEG [9], but the Swin-T's performance cannot be improved (being 81.2%).

## 4.2 Semantic Segmentation on ADE20K

We further evaluate the performance on segmentation tasks. We test on the ADE20K dataset [42], a challenging scene parsing task for semantic segmentation, which is popularly evaluated by recent Transformer-based methods. This dataset contains 20K images for training and 2K images for validation. Following the common practices, we use the training set to train our models and report the mIoU on the validation set. All models are pretrained on the ImageNet-1k dataset.

**Twins-PCPVT vs. PVT.** We compare our Twins-PCPVT with PVT [8] because they have similar design and computational complexity. To make fair comparisons, we use the Semantic FPN framework [43] and exactly the same training settings as in PVT. Specifically, we train 80K steps with a batch size of 16 using AdamW [37]. The learning rate is initialized as $1\times10^{-4}$ and scheduled by the 'poly' strategy with the power coefficient of 0.9. We apply the drop-path regularization of 0.2 for the backbone and weight decay 0.0005 for the whole network. Note that we use a stronger drop-path regularization of 0.4 for the large model to avoid over-fitting. For Swin, we use their official code and trained models. We report the results in Table 2. With comparable FLOPs, Twins-PCPVT-S outperforms PVT-Small with a large margin (+4.5% mIoU), which also surpasses ResNet-50 by 7.6% mIoU. It also outperforms Swin-T with a clear margin. Besides, Twins-PCPVT-B also achieves 3.3% higher mIoU than PVT-Medium, and Twins-PCPVT-L surpasses PVT-Large with 4.3% higher mIoU.

**Twins-SVT vs. Swin.** We also compare our Twins-SVT with the recent state-of-the-art model Swin [4]. With the Semantic FPN framework and the above settings, Twins-SVT-S achieves better performance (+1.7%) than Swin-T. Twins-SVT-B obtains comparable performance with Swin-S and Twins-SVT-L outperforms Swin-B by 0.7% mIoU (left columns in Table 2). In addition, Swin evaluates its performance using the UperNet framework [44]. We transfer our method to this framework and use exactly the same training settings as [4]. To be specific, we use the AdamW optimizer to train all models for 160k iterations with a global batch size of 16. The initial learning rate is $6\times10^{-5}$ and linearly decayed to zero. We also utilize warm-up during the first 1500 iterations. Moreover, we apply the drop-path regularization of 0.2 for the backbone and weight decay 0.01 for the whole network. We report the mIoU of both single scale and multi-scale testing (we use scales from 0.5 to 1.75 with step 0.25) in the right columns of Table 2. Both with multi-scale testing, Twins-SVT-S outperforms Swin-T by 1.3% mIoU. Moreover, Twins-SVT-L achieves new state of the art result 50.2% mIoU under comparable FLOPs and outperforms Swin-B by 0.5% mIoU. Twins-PCPVT also achieves comparable performance to Swin [4].

## 4.3 Object Detection and Segmentation on COCO

We evaluate the performance of our method using two representative frameworks: RetinaNet [46] and Mask RCNN [47]. Specifically, we use our transformer models to build the backbones of these detectors. All the models are trained under the same setting as in [8]. Since PVT and Swin report their results using different frameworks, we try to make fair comparison and build consistent settings for future methods. Specifically, we report standard $1\times$-schedule (12 epochs) detection results on the COCO 2017 dataset [48] in Tables 3 and 4. As for the evaluation based on RetinaNet, we train

**Table 1** – Comparisons with state-of-the-art methods for ImageNet-1K classification. Throughput is tested on the batch size of 192 on a single V100 GPU. All models are trained and evaluated on 224×224 resolution on ImageNet-1K dataset. [†]: w/ CPVT's position encodings [9].

| Method | Param (M) | FLOPs (G) | Throughput (Images/s) | Top-1 (%) |
|---|---|---|---|---|
| ConvNet | | | | |
| RegNetY-4G [40] | 21 | 4.0 | 1157 | 80.0 |
| RegNetY-8G [40] | 39 | 8.0 | 592 | 81.7 |
| RegNetY-16G [40] | 84 | 16.0 | 335 | 82.9 |
| Transformer | | | | |
| DeiT-Small/16 [2] | 22.1 | 4.6 | 437 | 79.9 |
| CrossViT-S [30] | 26.7 | 5.6 | - | 81.0 |
| T2T-ViT-14 [27] | 22 | 5.2 | - | 81.5 |
| TNT-S [15] | 23.8 | 5.2 | - | 81.3 |
| CoaT Mini [17] | 10 | 6.8 | - | 80.8 |
| CoaT-Lite Small [17] | 20 | 4.0 | - | 81.9 |
| PVT-Small [8] | 24.5 | 3.8 | 820 | 79.8 |
| CPVT-Small-GAP [9] | 23 | 4.6 | 817 | 81.5 |
| Twins-PCPVT-S (**ours**) | 24.1 | 3.8 | 815 | 81.2 (+1.3) |
| Swin-T [4] | 29 | 4.5 | 766 | 81.3 |
| Swin-T + CPVT[†] | 28 | 4.4 | 766 | 81.2 |
| Twins-SVT-S (**ours**) | 24 | 2.9 | 1059 | 81.7 (+1.8) |
| T2T-ViT-19 [27] | 39.2 | 8.9 | - | 81.9 |
| PVT-Medium [8] | 44.2 | 6.7 | 526 | 81.2 |
| Twins-PCPVT-B(**ours**) | 43.8 | 6.7 | 525 | 82.7 (+0.8) |
| Swin-S [4] | 50 | 8.7 | 444 | 83.0 |
| Twins-SVT-B (**ours**) | 56 | 8.6 | 469 | 83.2 (+1.3) |
| ViT-Base/16 [1] | 86.6 | 17.6 | 86 | 77.9 |
| DeiT-Base/16 [2] | 86.6 | 17.6 | 292 | 81.8 |
| T2T-ViT-24 [27] | 64.1 | 14.1 | - | 82.3 |
| CrossViT-B [30] | 104.7 | 21.2 | - | 82.2 |
| TNT-B [15] | 66 | 14.1 | - | 82.8 |
| CPVT-B [9] | 88 | 17.6 | 292 | 82.3 |
| PVT-Large [8] | 61.4 | 9.8 | 367 | 81.7 |
| Twins-PCPVT-L(**ours**) | 60.9 | 9.8 | 367 | 83.1 (+5.2) |
| Swin-B [4] | 88 | 15.4 | 275 | 83.3 |
| Twins-SVT-L (**ours**) | 99.2 | 15.1 | 288 | 83.7 (+5.8) |
| Hybrid | | | | |
| BoTNet-S1-59 [29] | 33.5 | 7.3 | - | 81.7 |
| BossNet-T1 [41] | - | 7.9 | - | 81.9 |
| CvT-13 [31] | 20 | 4.5 | - | 81.6 |
| BoTNet-S1-110 [29] | 54.7 | 10.9 | - | 82.8 |
| CvT-21 [31] | 32 | 7.1 | - | 82.5 |

all the models using AdamW [37] optimizer for 12 epochs with a batch size of 16. The initial learning rate is $1 \times 10^{-4}$, started with 500-iteration warmup and decayed by $10 \times$ at the 8th and 11th epoch, respectively. We use stochastic drop path regularization of 0.2 and weight decay 0.0001. The implementation is based on MMDetection [49]. For the Mask R-CNN framework, we use the initial learning rate of $2 \times 10^{-4}$ as in [8]. All other hyper-parameters follow the default settings in MMDetection. As for $3 \times$ experiments, we follow the common multi-scale training in [3, 4], *i.e.*, randomly resizing the input image so that its shorter side is between 480 and 800 while keeping longer one less than 1333. Moreover, for $3 \times$ training of Mask R-CNN, we use an initial learning rate of 0.0001 and weight decay of 0.05 for the whole network as [4].

For $1 \times$ schedule object detection with RetinaNet, Twins-PCPVT-S surpasses PVT-Small with 2.6% mAP and Twins-PCPVT-B exceeds PVT-Medium by 2.4% mAP on the COCO `val2017` split. Twins-SVT-S outperforms Swin-T with 1.5% mAP while using 12% fewer FLOPs. Our method outperform the others with similar advantage in $3 \times$ experiments.

Table 2 – Performance comparisons with different backbones on ADE20K validation dataset. FLOPs are tested on 512×512 resolution. All backbones are pretrained on ImageNet-1k except SETR [45], which is pretrained on ImageNet-21k dataset.

| Backbone | Semantic FPN 80k (PVT [8] setting) | | | Upernet 160k (Swin [4] setting) | | |
|---|---|---|---|---|---|---|
| | FLOPs (G) | Param (M) | mIoU (%) | FLOPs (G) | Param (M) | mIoU/MS mIoU (%) |
| ResNet50 [10] | 45 | 28.5 | 36.7 | - | - | - |
| PVT-Small [8] | 40 | 28.2 | 39.8 | - | - | - |
| Twins-PCPVT-S (ours) | 40 | 28.4 | 44.3 (+7.6) | 234 | 54.6 | 46.2/47.5 |
| Swin-T [4] | 46 | 31.9 | 41.5 | 237 | 59.9 | 44.5/45.8 |
| Twins-SVT-S (ours) | 37 | 28.3 | 43.2 (+6.5) | 228 | 54.4 | 46.2/47.1 |
| ResNet101 [10] | 66 | 47.5 | 38.8 | 258 | 86 | -/44.9 |
| PVT-Medium [8] | 55 | 48.0 | 41.6 | - | - | - |
| Twins-PCPVT-B (ours) | 55 | 48.1 | 44.9 (+6.1) | 250 | 74.3 | 47.1/48.4 |
| Swin-S [4] | 70 | 53.2 | 45.2 | 261 | 81.3 | 47.6/49.5 |
| Twins-SVT-B (ours) | 67 | 60.4 | 45.3 (+6.5) | 261 | 88.5 | 47.7/48.9 |
| ResNetXt101-64×4d [13] | - | 86.4 | 40.2 | - | - | - |
| PVT-Large [8] | 71 | 65.1 | 42.1 | - | - | - |
| Twins-PCPVT-L (ours) | 71 | 65.3 | 46.4 (+6.2) | 269 | 91.5 | 48.6/49.8 |
| Swin-B [4] | 107 | 91.2 | 46.0 | 299 | 121 | 48.1/49.7 |
| Twins-SVT-L (ours) | 102 | 103.7 | 46.7 (+6.5) | 297 | 133 | 48.8/50.2 |
| Backbone | PUP (SETR [45] setting) | | | MLA (SETR [45] setting) | | |
| T-Large (SETR) [45] | - | 310 | 50.1 | - | 308 | 48.6/50.3 |

For 1× object segmentation with the Mask R-CNN framework, Twins-PCPVT-S brings similar improvements (+2.5% mAP) over PVT-Small. Compared with PVT-Medium, Twins-PCPVT-B obtains 2.6% higher mAP, which is also on par with that of Swin. Both Twins-SVT-S and Twins-SVT-B achieve better or slightly better performance compared to the counterparts of Swin. As for large models, our results are shown in Table 1 (in supplementary) and we also achieve better performance with comparable FLOPs.

Table 3 – Object detection performance on the COCO `val2017` split using the RetinaNet framework. 1× is 12 epochs and 3× is 36 epochs. "MS": Multi-scale training. FLOPs are evaluated on 800×600 resolution.

| Backbone | FLOPs (G) | Param (M) | RetinaNet 1× | | | | | | RetinaNet 3× + MS | | | | | |
|---|---|---|---|---|---|---|---|---|---|---|---|---|---|---|
| | | | AP | $AP_{50}$ | $AP_{75}$ | $AP_S$ | $AP_M$ | $AP_L$ | AP | $AP_{50}$ | $AP_{75}$ | $AP_S$ | $AP_M$ | $AP_L$ |
| ResNet50 [10] | 111 | 37.7 | 36.3 | 55.3 | 38.6 | 19.3 | 40.0 | 48.8 | 39.0 | 58.4 | 41.8 | 22.4 | 42.8 | 51.6 |
| PVT-Small [8] | 118 | 34.2 | 40.4 | 61.3 | 43.0 | 25.0 | 42.9 | 55.7 | 42.2 | 62.7 | 45.0 | 26.2 | 45.2 | 57.2 |
| Twins-PCPVT-S (ours) | 118 | 34.4 | 43.0(+6.7) | 64.1 | 46.0 | 27.5 | 46.3 | 57.3 | 45.2(+6.2) | 66.5 | 48.6 | 30.0 | 48.8 | 58.9 |
| Swin-T [4] | 118 | 38.5 | 41.5 | 62.1 | 44.2 | 25.1 | 44.9 | 55.5 | 43.9 | 64.8 | 47.1 | 28.4 | 47.2 | 57.8 |
| Twins-SVT-S (ours) | 104 | 34.3 | 43.0(+6.7) | 64.2 | 46.3 | 28.0 | 46.4 | 57.5 | 45.6(+6.6) | 67.1 | 48.6 | 29.8 | 49.3 | 60.0 |
| ResNet101 [10] | 149 | 56.7 | 38.5 | 57.8 | 41.2 | 21.4 | 42.6 | 51.1 | 40.9 | 60.1 | 44.0 | 23.7 | 45.0 | 53.8 |
| ResNeXt101-32×4d [13] | 151 | 56.4 | 39.9 | 59.6 | 42.7 | 22.3 | 44.2 | 52.5 | 41.4 | 61.0 | 44.3 | 23.9 | 45.5 | 53.7 |
| PVT-Medium [8] | 151 | 53.9 | 41.9 | 63.1 | 44.3 | 25.0 | 44.9 | 57.6 | 43.2 | 63.8 | 46.1 | 27.3 | 46.3 | 58.9 |
| Twins-PCPVT-B (ours) | 151 | 54.1 | 44.3(+5.8) | 65.6 | 47.3 | 27.9 | 47.9 | 59.6 | 46.4(+5.5) | 67.7 | 49.8 | 31.3 | 50.2 | 61.4 |
| Swin-S [4] | 162 | 59.8 | 44.5 | 65.7 | 47.5 | 27.4 | 48.0 | 59.9 | 46.3 | 67.4 | 49.8 | 31.1 | 50.3 | 60.9 |
| Twins-SVT-B (ours) | 163 | 67.0 | 45.3(+6.8) | 66.7 | 48.1 | 28.5 | 48.9 | 60.6 | 46.9(+6.0) | 68.0 | 50.2 | 31.7 | 50.3 | 61.8 |

## 4.4 Ablation Studies

**Configurations of LSA and GSA blocks.** We evaluate different combinations of LSA and GSA based on our small model and present the ablation results in Table 5. The models with only locally-grouped attention fail to

Table 5 – Classification performance for different combinations of LSA (L) and GSA (G) blocks based on the small model.

| Function Type | Params (M) | FLOPs (G) | Top-1 (%) |
|---|---|---|---|
| (L, L, L) | 8.8 | 2.2 | 76.9 |
| (L, LLG, LLG, G) | 23.5 | 2.8 | 81.5 |
| (L, LG, LG, G) | 24.1 | 2.8 | 81.7 |
| (L, L, L, G) | 22.2 | 2.9 | 80.5 |
| PVT-small (G, G, G, G) [8] | 24.5 | 3.8 | 79.8 |

**Table 4** – Object detection and instance segmentation performance on the COCO `val2017` dataset using the Mask R-CNN framework. FLOPs are evaluated on a 800×600 image.

| Backbone | FLOPs (G) | Param (M) | Mask R-CNN 1× | | | | | | Mask R-CNN 3× + MS | | | | | |
|---|---|---|---|---|---|---|---|---|---|---|---|---|---|---|
| | | | $AP^b$ | $AP_{50}^b$ | $AP_{75}^b$ | $AP^m$ | $AP_{50}^m$ | $AP_{75}^m$ | $AP^b$ | $AP_{50}^b$ | $AP_{75}^b$ | $AP^m$ | $AP_{50}^m$ | $AP_{75}^m$ |
| ResNet50 [10] | 174 | 44.2 | 38.0 | 58.6 | 41.4 | 34.4 | 55.1 | 36.7 | 41.0 | 61.7 | 44.9 | 37.1 | 58.4 | 40.1 |
| PVT-Small [8] | 178 | 44.1 | 40.4 | 62.9 | 43.8 | 37.8 | 60.1 | 40.3 | 43.0 | 65.3 | 46.9 | 39.9 | 62.5 | 42.8 |
| Twins-PCPVT-S (ours) | 178 | 44.3 | $42.9_{(+4.9)}$ | 65.8 | 47.1 | $40.0_{(+5.6)}$ | 62.7 | 42.9 | $46.8_{(+5.8)}$ | 69.3 | 51.8 | 42.6 | 66.3 | 46.0 |
| Swin-T [4] | 177 | 47.8 | 42.2 | 64.6 | 46.2 | 39.1 | 61.6 | 42.0 | 46.0 | 68.2 | 50.2 | 41.6 | 65.1 | 44.8 |
| Twins-SVT-S (ours) | 164 | 44.0 | $43.4_{(+5.4)}$ | 66.0 | 47.3 | $40.3_{(+5.9)}$ | 63.2 | 43.4 | $46.8_{(+5.8)}$ | 69.2 | 51.2 | 42.6 | 66.3 | 45.8 |
| ResNet101 [10] | 210 | 63.2 | 40.4 | 61.1 | 44.2 | 36.4 | 57.7 | 38.8 | 42.8 | 63.2 | 47.1 | 38.5 | 60.1 | 41.3 |
| ResNeXt101-32×4d [13] | 212 | 62.8 | 41.9 | 62.5 | 45.9 | 37.5 | 59.4 | 40.2 | 44.0 | 64.4 | 48.0 | 39.2 | 61.4 | 41.9 |
| PVT-Medium [8] | 211 | 63.9 | 42.0 | 64.4 | 45.6 | 39.0 | 61.6 | 42.1 | 44.2 | 66.0 | 48.2 | 40.5 | 63.1 | 43.5 |
| Twins-PCPVT-B (ours) | 211 | 64.0 | $44.6_{(+4.2)}$ | 66.7 | 48.9 | $40.9_{(+4.5)}$ | 63.8 | 44.2 | $47.9_{(+5.1)}$ | 70.1 | 52.5 | 43.2 | 67.2 | 46.3 |
| Swin-S [4] | 222 | 69.1 | 44.8 | 66.6 | 48.9 | 40.9 | 63.4 | 44.2 | 47.6 | 69.4 | 52.5 | 42.8 | 66.5 | 46.4 |
| Twins-SVT-B (ours) | 224 | 76.3 | $45.2_{(+4.8)}$ | 67.6 | 49.3 | $41.5_{(+5.1)}$ | 64.5 | 44.8 | $48.0_{(+5.2)}$ | 69.5 | 52.7 | 43.0 | 66.8 | 46.6 |

obtain good performance (76.9%)
because this setting has a limited and small receptive field. An extra global attention layer in the last stage can improve the classification performance by 3.6%. Local-Local-Global (*abbr.* LLG) also achieves good performance (81.5%), but we do not use this design in this work.

**Sub-sampling functions.** We further study how the different sub-sampling functions affect the performance. Specifically, we compare the regular strided convolutions, separable convolutions and average pooling based on the 'small' model and present the results in Table 6. The first option performs best and therefore we choose it as our default implementation.

**Table 6** – ImageNet classification performance of different forms of sub-sampled functions for the global sub-sampled attention (GSA).

| Function Type | Top-1(%) |
|---|---|
| 2D Conv. | 81.7 |
| 2D Separable Conv. | 81.2 |
| Average Pooling | 81.2 |

**Positional Encodings.** We replace the relative positional encoding with CPVT for Swin-T and report the detection performance on COCO with RetinaNet and Mask R-CNN in Table 7. The CPVT-based Swin cannot achieve improved performance with both frameworks, which indicates that our performance improvements should be owing to the paradigm of Twins-SVT instead of the positional encodings.

**Table 7** – Object detection performance on the COCO using different positional encoding strategies.

| Backbone | RetinaNet | | | | | Mask RCNN | | | | |
|---|---|---|---|---|---|---|---|---|---|---|
| | FLOPs(G) | Param(M) | AP | $AP_{50}$ | $AP_{75}$ | FLOPs(G) | Param(M) | AP | $AP_{50}$ | $AP_{75}$ |
| Swin-T [4] | 245 | 38.5 | 41.5 | 62.1 | 44.2 | 264 | 47.8 | 42.2 | 64.6 | 46.2 |
| Swin-T+CPVT | 245 | 38.5 | 41.3 | 62.4 | 44.1 | 263 | 47.8 | 42.0 | 64.5 | 45.9 |

# 5 Conclusion

In this paper, we have presented two powerful vision transformer backbones for both image-level classification and a few downstream dense prediction tasks. We dub them as twin transformers: Twins-PCPVT and Twins-SVT. The former variant explores the applicability of conditional positional encodings [9] in pyramid vision transformer [8], confirming its potential for improving backbones in many vision tasks. In the latter variant we revisit current attention design to proffer a more efficient attention paradigm. We find that interleaving local and global attention can produce impressive results, yet it comes with higher throughputs. *Both transformer models set a new state of the art in image classification, objection detection and semantic/instance segmentation.*

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
