# Supplementary of "Twins: Revisiting the Design of Spatial Attention in Vision Transformer"

## A  Experiment

**Table 1** – Large models' object detection performance on the COCO `val2017` split using $1\times$ schedule.

| Backbone | RetinaNet $1\times$ | | | | Mask R-CNN $1\times$ | | | |
|---|---|---|---|---|---|---|---|---|
| | Param(M) | AP | $AP_{50}$ | $AP_{75}$ | FLOPs(G) | Param(M) | $AP^b$ | $AP^m$ |
| ResNeXt101-64$\times$4d [1] | 95.5 | 41.0 | 60.9 | 44.0 | 493 | 101.9 | 42.8 | 38.4 |
| PVT-Large [2] | 71.1 | 42.6 | 63.7 | 45.4 | 364 | 81.0 | 42.9 | 39.5 |
| Twins-PCPVT-L (ours) | 71.2 | 45.1 (+4.1) | 66.4 | 48.4 | 364 | 81.2 | 45.4 (+2.6) | 41.5 |
| Swin-B [3] | 98.4 | 44.7 | 65.9 | 47.8 | 496 | 107.2 | 45.5 | 41.3 |
| Twins-SVT-L (ours) | 110.9 | 45.7 (+4.7) | 67.1 | 49.2 | 474 | 119.7 | 45.9 (+3.1) | 41.6 |

## B  Algorithm

**Algorithm 1** PyTorch snippet of LSA.

```
class GroupAttention(nn.Module):
    def __init__(self, dim, num_heads=8, qkv_bias=False, qk_scale=None, attn_drop=0., proj_drop=0., k1=7,
        k2=7):
        super(GroupAttention, self).__init__()
        self.dim = dim
        self.num_heads = num_heads
        head_dim = dim // num_heads
        self.scale = qk_scale or head_dim ** -0.5
        self.qkv = nn.Linear(dim, dim * 3, bias=qkv_bias)
        self.attn_drop = nn.Dropout(attn_drop)
        self.proj = nn.Linear(dim, dim)
        self.proj_drop = nn.Dropout(proj_drop)
        self.k1 = k1
        self.k2 = k2

    def forward(self, x, H, W):
        B, N, C = x.shape
        h_group, w_group = H // self.k1, W // self.k2
        total_groups = h_group * w_group
        x = x.reshape(B, h_group, self.k1, w_group, self.k2, C).transpose(2, 3)
        qkv = self.qkv(x).reshape(B, total_groups, -1, 3, self.num_heads, C // self.num_heads).permute(3, 0,
            1, 4, 2, 5)
        q, k, v = qkv[0], qkv[1], qkv[2]
        attn = (q @ k.transpose(-2, -1)) * self.scale
        attn = attn.softmax(dim=-1)
        attn = self.attn_drop(attn)
        attn = (attn @ v).transpose(2, 3).reshape(B, h_group, w_group, self.k1, self.k2, C)
        x = attn.transpose(2, 3).reshape(B, N, C)
        x = self.proj(x)
        x = self.proj_drop(x)
        return x
```

35th Conference on Neural Information Processing Systems (NeurIPS 2021).

**Table 2** – Configuration details of Twins-PCPVT.

| | Output Size | Layer Name | Twins-PCPVT-S | Twins-PCPVT-B | Twins-PCPVT-L |
|---|---|---|---|---|---|
| Stage 1 | $\frac{H}{4} \times \frac{W}{4}$ | Patch Embedding | $P_1 = 4; C_1 = 64$ | | |
| | | Transformer Encoder with PEG | $\begin{bmatrix} R_1 = 8 \\ N_1 = 1 \\ E_1 = 8 \end{bmatrix} \times 3$ | $\begin{bmatrix} R_1 = 8 \\ N_1 = 1 \\ E_1 = 8 \end{bmatrix} \times 3$ | $\begin{bmatrix} R_1 = 8 \\ N_1 = 1 \\ E_1 = 8 \end{bmatrix} \times 3$ |
| Stage 2 | $\frac{H}{8} \times \frac{W}{8}$ | Patch Embedding | $P_2 = 2; C_2 = 128$ | | |
| | | Transformer Encoder with PEG | $\begin{bmatrix} R_2 = 4 \\ N_2 = 2 \\ E_2 = 8 \end{bmatrix} \times 3$ | $\begin{bmatrix} R_2 = 4 \\ N_2 = 2 \\ E_2 = 8 \end{bmatrix} \times 3$ | $\begin{bmatrix} R_2 = 4 \\ N_2 = 2 \\ E_2 = 8 \end{bmatrix} \times 8$ |
| Stage 3 | $\frac{H}{16} \times \frac{W}{16}$ | Patch Embedding | $P_3 = 2; C_3 = 320$ | | |
| | | Transformer Encoder with PEG | $\begin{bmatrix} R_3 = 2 \\ N_3 = 5 \\ E_3 = 4 \end{bmatrix} \times 6$ | $\begin{bmatrix} R_3 = 2 \\ N_3 = 5 \\ E_3 = 4 \end{bmatrix} \times 18$ | $\begin{bmatrix} R_3 = 2 \\ N_3 = 5 \\ E_3 = 4 \end{bmatrix} \times 27$ |
| Stage 4 | $\frac{H}{32} \times \frac{W}{32}$ | Patch Embedding | $P_4 = 2; C_4 = 512$ | | |
| | | Transformer Encoder with PEG | $\begin{bmatrix} R_4 = 1 \\ N_4 = 8 \\ E_4 = 4 \end{bmatrix} \times 3$ | $\begin{bmatrix} R_4 = 1 \\ N_4 = 8 \\ E_4 = 4 \end{bmatrix} \times 3$ | $\begin{bmatrix} R_4 = 1 \\ N_4 = 8 \\ E_4 = 4 \end{bmatrix} \times 3$ |

**Table 3** – Configuration details of Twins-SVT.

| | Output Size | Layer Name | Twins-SVT-S | Twins-SVT-B | Twins-SVT-L |
|---|---|---|---|---|---|
| Stage 1 | $\frac{H}{4} \times \frac{W}{4}$ | Patch Embedding | $P_1 = 4; C_1 = 64$ | $P_1 = 4; C_1 = 96$ | $P_1 = 4; C_1 = 128$ |
| | | Transformer Encoder w/ PEG | $\begin{bmatrix} LSA \\ GSA \end{bmatrix} \times 1$ | $\begin{bmatrix} LSA \\ GSA \end{bmatrix} \times 1$ | $\begin{bmatrix} LSA \\ GSA \end{bmatrix} \times 1$ |
| Stage 2 | $\frac{H}{8} \times \frac{W}{8}$ | Patch Embedding | $P_2 = 2; C_2 = 128$ | $P_2 = 2; C_2 = 192$ | $P_2 = 2; C_2 = 256$ |
| | | Transformer Encoder w/ PEG | $\begin{bmatrix} LSA \\ GSA \end{bmatrix} \times 1$ | $\begin{bmatrix} LSA \\ GSA \end{bmatrix} \times 1$ | $\begin{bmatrix} LSA \\ GSA \end{bmatrix} \times 1$ |
| Stage 3 | $\frac{H}{16} \times \frac{W}{16}$ | Patch Embedding | $P_3 = 2; C_3 = 256$ | $P_3 = 2; C_3 = 384$ | $P_3 = 2; C_3 = 512$ |
| | | Transformer Encoder w/ PEG | $\begin{bmatrix} LSA \\ GSA \end{bmatrix} \times 5$ | $\begin{bmatrix} LSA \\ GSA \end{bmatrix} \times 9$ | $\begin{bmatrix} LSA \\ GSA \end{bmatrix} \times 9$ |
| Stage 4 | $\frac{H}{32} \times \frac{W}{32}$ | Patch Embedding | $P_4 = 2; C_4 = 512$ | $P_4 = 2; C_4 = 768$ | $P_4 = 2; C_4 = 1024$ |
| | | Transformer Encoder w/ PEG | $\begin{bmatrix} GSA \end{bmatrix} \times 4$ | $\begin{bmatrix} GSA \end{bmatrix} \times 2$ | $\begin{bmatrix} GSA \end{bmatrix} \times 2$ |

# C  Architecture Setting