# OpenReview forum: "Twins: Revisiting the Design of Spatial Attention in Vision Transformers"
_NeurIPS.cc/2021/Conference — NeurIPS 2021 Poster_

### Official Review · Reviewer_uA3x · 2021-07-14

**Rating:** 6
**Confidence:** 3

**Summary:**

The paper proposes two new transformer architectures to address image classification and object detection and segmentation tasks, focusing on the attention mechanisms in each. The first method, Twins-PCPVT, combines the pyramidal subsampling of attention in PVT with conditional positional encoding of CPVT, the second, Twins-SVT, proposes to combine local grouping to manage computational complexity in attention with global subsampling to improve receptive field size, drawing analogies to separable filters.

Both methods are empirically analyzed on relevant datasets (Imagenet1k, MSCOCO, ADE20k) in comparison to state of the art methods (PVT, CPVT, Swin) and show improvements in accuracy and run-time speed.



**Limitations And Societal Impact:**

The authors do not discuss societal impact, but I do not have any specific concerns.

The authors do not speculate strongly on limitations of the work.

**Main Review:**

I am inclined to rate the paper marginally below the acceptance threshold based on the following analysis:

Strengths:
- The paper addresses an important problem: improving speed and accuracy of transformer architectures by designing attention mechanisms
- The empirical analysis appears thorough and on a wide set of relevant problems and data
- The methods are motivated reasonably well at the coarse level (combine subsampling with conditional encoding, combine local and global grouping, but see the concerns below)
- The empirical analysis shows numerical benefits in accuracy and speed across the tested models and data

Weakness:
- The novelty of the methods and what sets them apart from the state of the art is not clearly highlighted and (at least for me) needed several re-reads. The paper would benefit from outlining more clearly what the novelty and contribution of the paper are.
- It is at times hard to follow the exposure: (a) Some terminology is not explained: Why are the methods termed "Twins", why is one PCPVT, the other SVT (l. 96ff)? The reader is left to interpret. (b) The two methods appear valid on their own, but I do not see unification (l39) beyond addressing attention mechanisms. If the authors see more arguments that the paper leads to a more coherent picture of attention mechanisms, they should point this out more clearly. As it is, the title, abstract (ll. 3-5) and introduction (ll. 39ff) left me with the impression that the paper was aiming towards coherence or unification of attention methods, but I saw little of this evidenced in the remainder of the paper.
- While the high-level motivation appears sound (as pointed out above), the motivation for the individual choices of implementation in the Twins-SVT method (section 3.2) are not clear to me. I would like to understand why these particular choices are suitable, what are other options and why where they not chosen. I understand that the paper is primarily geared towards empirical evidence (versus theoretical motivation and analysis), but I feel that I am not learning much about why and if this is a good choice beyond the empirics.

Further comments/suggestions, these do not impact the review rating:
- I feel that drawing the analogies to separable convolutions or separable filtering in general does not add to the paper. The motivation to bridge fine-grained, local and coarse, global information is fine by itself and does not need the motivation from separable convolutions.
- The authors indicate deployment frameworks such as ONNX and TensorRT as a motivation from the runtime speed perspective. It would be nice to demonstrated improvements there directly (by showing feasibility and/or runtime improvements at deployment) to substantiate this motivation.


[Discussion period update]
I appreciate that the authors added some details based on my concerns during the review phase. I feel that my initial concerns on outlining the novelty more directly and clarifying "unification" and some of the terminology are reasonably well addressed. My concern on the largely empirical choices in 3.2 is dampened, but not removed. Notwithstanding this, I feel compelled to upgrade my rating.

**Time Spent Reviewing:**

12

---

> ### Author Response · Authors · 2021-08-09
> **Global-global and local-global are both good design patterns for vision transformers**
>
> Thanks for spending your precious time on the review.
>
> $\textbf{Q1:}$ The paper would benefit from outlining more clearly what the novelty and contribution of the paper are.
>
> $\textbf{A1:}$  We will make it clear in the updated version. Maybe some more backgrounds will help here.
>
> Most recently, Swin is the new state-of-the-art Vision Transformer that outperforms previous notable works (e.g., DeiT, PVT) by clear margins. Noticeably, Swin is based on the alternation design of local windows and shifted ones (`local-shifted local`),  However, there are two basic questions to be answered. One is whether the `global-global` design (as in PVT) is inherently worse than the `local-shifted local' design in Swin. Another is whether there are other alternatives comparable to Swin, especially observing that the shifted window of Swin is somewhat complicated and harder to deploy (see $\textbf{A1}$ to Reviewer oWv9). Therefore, in our paper, we focus on the new, simpler architectural design, which achieves on par results without these drawbacks.
>
> In a nutshell, we propose two types of simple and competitive models, Twins-PCPVT and Twins-SVT. The former inherits a `global-global` attention design from PVT, where we unveil that PVT is inferior to Swin mainly because it previously uses an improper positional encoding strategy. We replace the PE with PEG (from CPVT) and propose Twins-PCPVT to resolve this drawback.
>
> Besides, we present Twins-SVT, which is an alternation of the local and global attention design (`local-global`). Not only does Twins-SVT greatly reduce the computational cost of the global attention, but it also delivers improved performance than Twins-PCPVT.
>
> $\textbf{Q2:}$ It is at times hard to follow the exposure. (a) The naming of Twins, PCVPT, and SVT. (b)  The two methods appear valid on their own, but not show unification beyond addressing attention mechanisms.
>
> $\textbf{A2:}$
>  (a) We name our method "Twins" because we introduce two simple yet high-performance vision transformer designs (one with `global-global` attention and the other alternating `local-global` attention). They are both comparable to the state-of-the-art Swin Transformer. As mentioned in Line 95-98, PCPVT is named after PVT+CPVT and means *Pyramid CPVT*. SVT is short for *Spatially-separable Vision Transformer*.
> (b) Our paper revisits the short and long spatial attention mechanism in vision transformers. The unified perspective refers to the attention mechanisms, for example, `global-global` or `local-global`. Both require delicate manipulation to make them effective and efficient.
>
> $\textbf{Q3:}$ The motivation for the individual choices of implementation in the Twins-SVT method (Section 3.2) is not clear to me.
>
> $\textbf{A3:}$ Our core motivation of Twins-SVT models is to use an alternation of local attention and global attention while keeping a small number of parameters and low FLOPs in mind. This design largely reduces the computational complexity, yet still has the global receptive field, thus retaining good performance.
>
> $\textbf{Q4:}$ It would be nice to demonstrated improvements there directly (by showing the feasibility and/or runtime improvements at deployment.
>
> $\textbf{A4:}$ Thanks for this suggestion, and we will add it to the final version. Twins models enjoy $\sim$1.6 times speedup on NVIDIA TensorRT 7.0, from $1059$ images/s to $1732$ (Twins-SVT-S).  In contrast,  Swin is not TensorRT-exportable and has a PyTorch-throughput of $766$ images/s (see Table 1).

---

> > ### Author Response · Authors · 2021-08-26
> > **Have we addressed your concerns?**
> >
> > Dear Reviewer uA3x,
> >
> > Hello, as the discussion period is running around the corner, we are eager to know whether our response addresses your concerns well. If you still have further specific questions, we would like to do our best to resolve them.
> >
> > Best regards,
> >
> > Paper774 Authors

---

> > > ### Author Response · Authors · 2021-08-29
> > > **Guidelines for designing SVT**
> > >
> > > Thanks for the further valuable feedback. Here we provide a perspective beyond the experimental results for SVT and hope it helps better explain why we make such choices and design.
> > >
> > > We have two basic guidelines for designing SVT: *a better trade-off between performance and FLOPS*, as well as *whether it is easy for deployment*. The latter requirement greatly reduces the design space because we only use well-supported and optimized operations across the most popular deployment frameworks.
> > >
> > > - **The design of local attention**. We make use of the locality prior of a 2D image instead of viewing it as a flattened sequence as in ViT. Self-attention is performed within each local group, thus avoiding the quadratically increasing  FLOPS of self-attention on the whole sequence (cf. line 140-144). This operation can be implemented with well-supported reshaping and matrix multiplication.
> > > - **The design of global attention**. Global attention here is the mechanism to communicate information among different groups. This is to make use of the well-known non-local advantage of the Transformer. To reduce the FLOPS of global attention across the whole sequence, we use GSA to sub-sample the keys and compute the attention. This can also be implemented with matrix multiplication and the sub-sampling function (we've tested strided convolution, depth-wise convolution, and average pooling here and they are well-supported by almost all frameworks, Table~6).
> > > - **Total FLOPS**. To optimize the above two components altogether, we calculate the optimal settings for local window sizes (c.f. line 163-174).

---

> > > > ### Comment · Reviewer_uA3x · 2021-08-31
> > > > **On empirical motivation and substantiation**
> > > >
> > > > Thank you for iterating over my concerns. I do appreciate the approach of addressing real-world factors in making design decisions ("a better trade-off between performance and FLOPS, as well as whether it is easy for deployment."). My concern is that this opens a whole new set of questions, such as (a) are FLOPS a good measurement of deployed performance (vs. other factors such as cache size, activation size in memory, dependencies between layers, compiler optimizations), (b) the supportability for easy deployment may vastly differ between end devices (from FPGA/ASIC over microcontroller to full-blown DL accelerator) that are out of focus of this paper. As an example, the mentioned depth-wise convolutions can pose a challenge versus full convolutions when quantizing to 8 bits or below (a significant speed-up on its own if the hardware supports it). The substantiation demonstrated in the context of the paper ("maps to matrix multiplication and sub-sampling") is sufficient as it is for a 6 rating, as the submission is not strongly focused on deployment, but uses the ease of deployment and speed improvements as justification for the design choices.

---

> > > > > ### Author Response · Authors · 2021-09-08
> > > > > **Easy-to-obtain and widely-used metrics are chosen for generality**
> > > > >
> > > > > Thanks for the timely response. For now, Twins by itself is a further step towards real-world applications for transformer-based architectures. Just as you mentioned, the deployed performance indeed involves many factors, some of which are hard to quantitatively evaluate and can vary constantly. Therefore, we try to design our model with some widely-used and easily available metrics including inference latency, throughputs, and FLOPS. Accurately modeling other possible factors is indeed out of the focus of this paper, we will consider them in our future research.

---

### Official Review · Reviewer_bc7u · 2021-07-16

**Rating:** 6
**Confidence:** 4

**Summary:**

This paper proposes to combine CPE with PVT. In addition, local attention is combined with a subsampled global attention layer so that the model can capture both local details and global relations efficiently. The design is verified on image classification, detection and segmentation.

**Limitations And Societal Impact:**

Yes.

**Main Review:**


## Originality
The Proposed Twins-PCPVT simply combines CPE with PVT in each pyramid resolution. This lacks enough novelty, except verifying the effectiveness of CPE. The Twins-SVT combines local windows attention from Swin and global attention from PVT on low resolution feature maps, but this is a relatively more interesting combination.

## Strength
1. The experiments are extensive, on three large scale datasets and tasks.
2. The proposed method is simple and intuitive.

## Weakness
1. Table 5 lacks enough explanation. Does (L, L, L) mean three stages only? In addition, (L, LLG, LLG, G) has fewer parameters than (L, LG, LG, G). Does it mean the total number of layers is fixed?
2. Another important baseline lacks explanation. In table 7, Swin is combined with CPVT, but is it applied to all stages as in Twins, or in the first stage only? What if the relative positional encoding is not removed?
3. The proposed method seems to require strong regularization, e.g. a larger stochastic depth rate. The strong regularization might be due to the small spatial resolution the extra global attention layers are applied. In addition, the modified gradient clipping is claimed as especially important but lacks enough explanation. Is it also due to the extra global layer?
4. It is not clear where the block number (in appendix table 3) comes from. For example, in Twins-SVT-S, 4 blocks are used for global attention. Is it modified from Swin for best result on ImageNet or downstream tasks?
5. Controlled comparison (where only one module is added or removed) is not sufficient for the main Twins-SVT contribution. Although the number of FLOPS and parameters are controlled, there are always more than one difference between compared models. Maybe one could first insert all the PEGs to Swin, then add the global attention PVT layers, and then probably show that shifted window is not necessary with the global attention layers.

### Typos and suggestions
1. Line 252, 160k iterations
2. Line 303, cannot achieve
3. Line 163, "key", change to important so that it is not confused with "attention key"?
4. Maybe emphasize that query is not downsampled in the global attention.

------------------------ Post Rebuttal ------------------------

Thanks the authors for the responses. The rebuttal addresses most of my concern regarding the clarity of the paper and the fair ablation/comparison with Swin, CPE, and RPE. So I tend to keep my original rating of 6.

**Time Spent Reviewing:**

4

---

> ### Author Response · Authors · 2021-08-09
> **Experiments are controlled with the best effort**
>
> Thanks for the deliberate examination.
>
> $\textbf{Q1:}$ Does (L, L, L) mean three stages only?  Does it mean the total number of layers is fixed?
>
> $\textbf{A1:}$ (1) Yes. (L, L, L) means three stages.
> (2) Yes, all other models except (L, L, L) in Table 5 have a fixed number of layers. We will make it clearer.
>
> Moreover, we need to note that (L, L, L, G) in Table 5 is equivalent to (L, L, L, L). This is because the common input shape of ImageNet classification is $224\times224$, and thus the spatial shape of the last stage is $7\times7$. As we use a local window size $7\times7$ in LSA, L in the last layer is equivalent to G. Therefore, we cannot have a four-stage model without global attention. This is why we design the three-stage baseline (L, L, L) to evaluate the performance where all attentions are local (c.f.,  Line 289).
>
> $\textbf{Q2:}$ Is it applied to all stages as in Twins, or in the first stage only? What if the relative positional encoding is not removed?
>
> $\textbf{A2:}$ For a fair comparison, CPVT's conditional position encodings (CPE) are applied to all stages of Swin just like Twins. Relative positional encoding (RPE) doesn't give much gain if we already have CPE. If we use both positional encodings in Swin, the object detection performance on COCO is nearly the same (changes are within $0.1$%-$0.2$% AP). In short, we have that Swin + CPE $\approx$ Swin + CPE + RPE. We will make this clearer in Table 7.
>
> $\textbf{Q3:}$ Twins seem to require strong regularization. In addition, the modified gradient clipping is claimed as especially important but lacks enough explanation. Is it also due to the extra global layer?
>
> $\textbf{A3:}$ We borrowed our training settings mostly from Swin, mainly for a fair comparison, and we did not use stronger regularization. Note that we use the same gradient clipping norm (i.e., 5) as in Swin's code. We meant that it is especially important for training large vision transformers in general, not just our models. Therefore, these regularization techniques are used not due to the global attention layers. We will clarify this point in a revision.
>
> $\textbf{Q4:}$ It is not clear where the block number (in Table 3 of the appendix) comes from. For example, in Twins-SVT-S, 4 blocks are used for global attention. Is it modified from Swin for the best result on ImageNet or downstream tasks?
>
> $\textbf{A4:}$ Thanks for noticing this subtle difference. The number of blocks of Twins-SVT-S is adjusted so that it can have a comparable number of parameters to both Swin-T and PVT-Small. We've also trained another model (though not listed in the paper) using the same number of blocks and channels as Swin-T, where we achieve $81.6$% top-1 accuracy on ImageNet. This result is very close to Twins-SVT-S in Table 1 and is still better than Swin. We keep the same number of blocks and channels for larger Twins-SVT models just as their Swin counterparts.
>
> $\textbf{Q5:}$ Controlled comparison (where only one module is added or removed) is not sufficient for the main Twins-SVT contribution. ... probably show that shifted window is not necessary with the global attention layers.
>
> $\textbf{A5:}$ As mentioned in $\textbf{A2}$ and shown in Table 1, 3, 4, and 7, the performance relationship is Twins + CPE $>$ Swin + CPE $\approx$ Swin + CPE + RPE. Note that the architecture settings of Twins-SVT-B and Twins-SVT-L (Table 3 in the supplementary) strictly follow Swin-S and Swin-B (see Table 7 in Swin's paper) respectively to make controlled comparisons. This setting purpose is shown in Line 188-190. In our paper,  we compare Twins-SVT-B with Swin-S across all benchmarks (Table 1,2,3,4). And Twins-SVT-L is compared with Swin-B in the same way. Based on this, we can claim Twins-SVT compares favorably with Swin. We don't mean that the shifted window is unnecessary. Instead, we aim to provide an alternative to the shifted window paradigm in Swin, while having higher performance, being simpler, and more efficient.
>
> We will fix the typos in the final version as suggested.

---

> > ### Author Response · Authors · 2021-08-30
> > **Are the previous concerns resolved?**
> >
> > Dear Reviewer bc7u,
> >
> > Hello,  on seeing that the discussion period is almost at the end, we are anxious to know whether our reply fairly addresses your concerns. We would like to do our best to resolve any further uncleared questions.
> >
> > Sincerely,
> >
> > Paper774 Authors

---

### Official Review · Reviewer_oWv9 · 2021-07-16

**Rating:** 7
**Confidence:** 5

**Summary:**

This paper presents a new transformer architecture for various vision tasks. The authors revisit the spatial attention design and propose integrating LSA(local self-attention) and GSA(global self-attention) for effective feature modeling. The experiment results have demonstrated the effectiveness of the Twins on many vision tasks, including image-level classification and dense prediction.

**Limitations And Societal Impact:**

#193~#199 has figured out the difference with Swin-transofmer. But I wonder why the authors claim the Swin-transformer is more complicated and cannot be optimized for speed on devices? Can the authors provide the evidence to support your claim? And in my opinion, the shifted window in Swin-transformer is also a straightforward idea, resulting in a simple but effective design.

**Main Review:**

1. The paper is easy to follow.
2. The idea of this paper is simple but effective. The author proposes to decompose global attention into two separate steps: firstly, applying local self-attention to aggregate features like the swin-transformer. Next, they propose using the summarized key to provide long-range information with low computation costs.  The most important thing of designing a transformer backbone is to alleviate the learning difficulty for the original global transformer, and the success of convolutional-based backbone gives the prior that the local interaction on features can speed up the backbone training. The idea of this paper has incorporated the design of the local window prior in convolution but uses a simple way to obtain long-range information by cross-attention. It is very cool, and I appreciate this idea.
3. The experiments on several tasks has achieved the state-of-the-art performance, and the ablation studies give detailed analysis on the key components of this paper

**Time Spent Reviewing:**

3 hours

---

> ### Author Response · Authors · 2021-08-09
> **Swin adds extra cost and is not easily exportable**
>
> Thanks for your considerate reviews.
>
> $\textbf{Q1:}$  I wonder why the authors claim the Swin-transformer is more complicated and cannot be optimized for speed on devices? Can the authors provide the evidence to support your claim? The shifted window in Swin-transformer is also a straightforward idea, resulting in a simple but effective design.
>
> $\textbf{A1:}$ Although the shifted window idea is straightforward, its implementation is not. Swin Transformer depends on `torch.roll()` to perform cyclic shift and its reverse on features. This operation is memory unfriendly and rarely supported by popular inference frameworks such as NVIDIA TensorRT, Google Tensorflow-Lite, and Snapdragon Neural Processing Engine SDK (SNPE), etc. This hinders the deployment of Swin either on the server-side or on end devices.
>
> In contrast, Twins models don't require such an operation and only involve matrix multiplications that are already optimized well in modern deep learning frameworks (Line 8).  Therefore, Twins can further benefit from the optimization in a production environment. For example, we converted Twins-SVT-S from PyTorch to TensorRT (v7.0),  and its throughput is boosted from $1059$ to $1732$ (measured in images/s, tested on Tesla V100). Other models of Twins also enjoy similar speed-ups ($1.6\times$).

---

> > ### Comment · Reviewer_oWv9 · 2021-09-06
> > **Final decision**
> >
> > I have carefully read the answers and other reviewers' feedback. I appreciate the idea, and the authors have addressed my concerns. I will keep my initial rating and support the paper to be accepted.

---

### Decision · Program_Chairs · 2021-09-27

**Decision:**

Accept (Poster)

**Comment:**

This submission received 3 positive final ratings: 7, 6, 6.
On the positive side, reviewers appreciated simplicity and effectiveness of the idea, strong empirical performance and clear presentation.
At the same time, some of them initially expressed concerns with overall novelty and motivation of certain design choices.
After an extensive discussion between the authors and the reviewers during the rebuttal period, one of the reviewers upgraded their score (from negative to positive), while others remained unchanged.
Overall, the strengths of this paper outweigh its weaknesses, so the final recommendation is to accept for poster presentation.